# Targeted memory reactivation in REM but not SWS selectively reduces arousal responses

Isabel C. Hutchison[1], Stefania Pezzoli [2,6,7], Maria-Efstratia Tsimpanouli [3], Mahmoud E. A. Abdellahi[4], Gorana Pobric[5], Johann Hulleman[5] & Penelope A. Lewis [4,5✉]

A growing body of evidence suggests that sleep can help to decouple the memory of emotional experiences from their associated affective charge. This process is thought to rely on the spontaneous reactivation of emotional memories during sleep, though it is still unclear which sleep stage is optimal for such reactivation. We examined this question by explicitly manipulating memory reactivation in both rapid-eye movement sleep (REM) and slow-wave sleep (SWS) using targeted memory reactivation (TMR) and testing the impact of this manipulation on habituation of subjective arousal responses across a night. Our results show that TMR during REM, but not SWS significantly decreased subjective arousal, and this effect is driven by the more negative stimuli. These results support one aspect of the sleep to forget, sleep to remember (SFSR) hypothesis which proposes that emotional memory reactivation during REM sleep underlies sleep-dependent habituation.

[1] Department of Neurology, Northwestern University, Chicago, IL, USA. [2] Department of Neuroscience, University of Sheffield, Sheffield, UK. [3] Department of Neurology and Sleep Disorders Center, University of Michigan, Ann Arbor, MI, USA. [4] School of Psychology, Cardiff University, Cardiff, UK. [5] Division of Neuroscience & Experimental Psychology, University of Manchester, Manchester, UK. [6] Present address: Molecular Biophysics and Integrated Bioimaging, Lawrence Berkeley National Laboratory, Berkeley, CA, USA. [7] Present address: Helen Wills Neuroscience Institute, University of California Berkeley, Berkeley, CA, USA. ✉email: lewisp8@cardiff.ac.uk

Sleep plays a crucial role in the consolidation of recently acquired memories[1]. However, not all memories are consolidated equally—experiences that elicit an emotional response tend to be remembered better across sleep than those that do not[2−5]. Furthermore, the emotional charge associated with these memories is gradually lost over time[6]. In the sleep to forget, sleep to remember (SFSR) hypothesis, Walker and colleagues propose that this dissipation of emotional charge relies on the reactivation of the emotional memory during rapid-eye movement (REM) sleep[7,8]. This proposal is supported by the observation that emotional habituation, which can be defined as a reduction in emotional response, occurs across periods of sleep containing REM[9]. REM has also been associated with overnight reductions in the extent to which the amygdala responds to emotional stimuli[8]. However, contrasting results suggest that emotional reactivity is maintained across REM[10,11], and there is also evidence for a more central role of slow-wave sleep (SWS) in emotional habituation[12,13].

Targeted memory reactivation (TMR) is a technique in which memory reactivation is intentionally triggered in sleep through the re-presentation of cues that were linked to the memory in the wake, and is commonly achieved using sounds or smells, see[14] for a review. While early studies reported a benefit of TMR in REM sleep on overnight memory consolidation[15,16], a growing number of reports suggest that TMR is beneficial when presented during non-REM[17−21], but not during REM[17,19,22] see[23] for meta-analysis. In terms of how TMR may influence emotional arousal, one recent study showed an impact of non-REM TMR on pleasantness and arousal ratings, but this only emerged in socially anxious participants, and after a week of consolidation[24]. Very few studies have investigated the impact of REM TMR on emotional material. Thus, inducing reactivation of fear memories during REM was shown to result in increased generalization[25], and TMR of a Pavlovian conditioning task during REM lead to increased habituation compared to TMR of the same task during stage 2 of non-REM[26].

Building on this literature, we set out to test the SFSR prediction that replaying emotionally arousing memories during REM, but not SWS, would be associated with a reduced arousal rating for this material the next day. We did this by manipulating the reactivation of emotional memories during sleep using TMR. Our participants rated emotionally negative and neutral picture-sound pairs for arousal both before and after a night of sleep. During sleep, we cued half of the negative and half of the neutral stimuli for reactivation by softly replaying the associated sounds. We then examined the impact of this TMR upon overnight habituation of the arousal response. We carefully controlled the sleep stage in which TMR was applied, cueing participants either in the REM (REM Group) or SWS (SWS Group), see Fig. 1. Based on the SFSR hypothesis, we predicted that TMR would lead to greater habituation when applied during REM, but not SWS.

## Results

At baseline, arousal ratings were higher for negative than for neutral items, showing that participants rated the stimuli in keeping with expectations. This was confirmed by a $2 \times 2 \times 2$ ANOVA on pre-sleep arousal with the factors Group, Cueing, and Emotion ($F_{(1,32)} = 337.93$, $p < 0.001$; Paired $t$ test $p < 0.001$ in both SWS and REM groups). Ratings did not differ between Cued and Un-cued stimuli prior to sleep (paired $t$ tests $p > 0.2$ in both cases), showing that the baseline was well balanced, and no participants reported awareness that sounds were played during their sleep.

We investigated how overnight habituation of arousal was modulated by TMR cueing using a $2 \times 2 \times 2$ ANOVA with factors Group, Cueing, and Emotion, see Table 1. Overnight habituation was calculated as (second pre-sleep arousal rating – post-sleep arousal rating) / first pre-sleep arousal rating. This revealed a

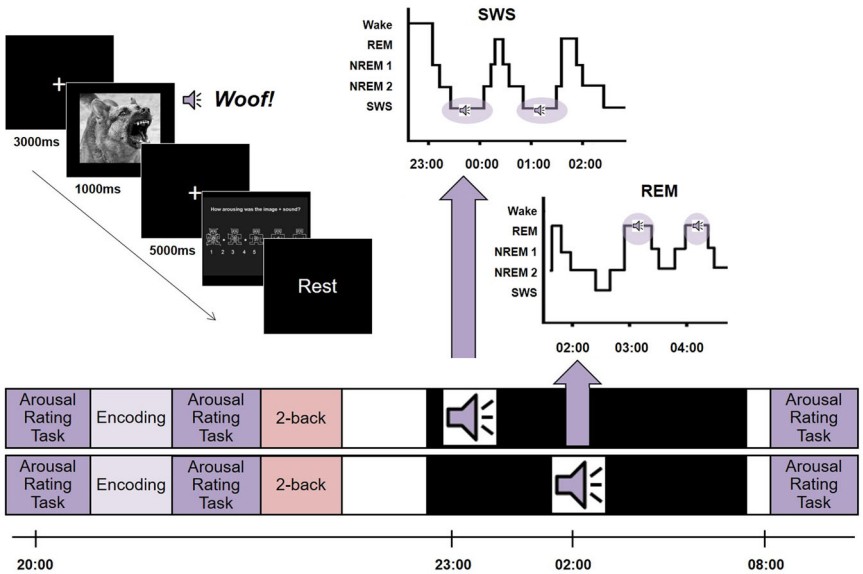

**Fig. 1 Experimental design.** In the evening session, participants rated each combined picture/sound stimulus for arousal. Next, they completed an encoding task in which they had to indicate which sound was paired with each image. They then completed the arousal rating a second time, as ratings may have changed as a result of the encoding task. Finally, participants performed a visual 2-back task using images that were not seen elsewhere in the experiment to create a buffer between exposure to the highly emotional stimuli and subsequent sleep. They were then fitted with electrodes for PSG recording and allowed to relax until lights out at approximately midnight. During the night, we performed TMR by exposing participants to the sounds associated with half of the neutral and half the negative learned stimuli in pseudorandom order five times each but avoiding adjacent repetitions. Sound cues were presented during REM in the REM group and during SWS in the SWS group. In the morning, participants were allowed thirty minutes to overcome sleep inertia before performing a final arousal rating task. We examined overnight change in arousal ratings to determine how these had changed across the night.

**Table 1 Average subjective arousal ratings at each session.**

| | | | First exposure | Pre-sleep exposure | Post-sleep exposure |
|---|---|---|---|---|---|
| REM group | Neutral | Cued | 2.70 ± 0.83 | 2.57 ± 0.86 | 2.39 ± 0.94 |
| | | Un-cued | 2.75 ± 0.97 | 2.62 ± 1.16 | 2.54 ± 1.23 |
| | Negative | Cued | 6.02 ± 1.62 | 6.20 ± 1.73 | 5.68 ± 1.71 |
| | | Un-cued | 6.20 ± 1.53 | 6.09 ± 1.64 | 5.86 ± 1.68 |
| SWS group | Neutral | Cued | 2.38 ± 1.01 | 2.27 ± 0.79 | 2.10 ± 0.64 |
| | | Un-cued | 2.33 ± 0.81 | 2.22 ± 0.72 | 2.10 ± 0.64 |
| | Negative | Cued | 5.62 ± 1.64 | 5.40 ± 1.54 | 5.34 ± 1.58 |
| | | Un-cued | 5.52 ± 1.57 | 5.47 ± 1.59 | 5.45 ± 1.66 |

Average ratings for neutral and negative, Un-cued and Cued stimuli at each of the three arousal rating sessions. These sessions occurred at the very start of the experiment, prior to 2-back task and half an hour after sleep, respectively. Average ratings are shown ± standard deviation. *REM* rapid-eye movement sleep, *SWS* slow-wave sleep. REM group, N = 15, SWS group, N = 18.

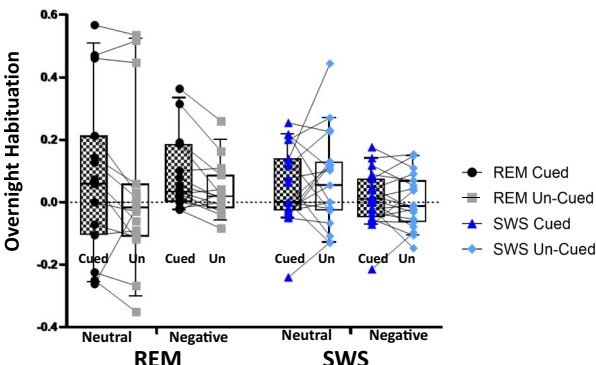

**Fig. 2 Overnight habituation of subjective arousal.** Cueing (black and white checked boxes) was associated with a significantly greater overnight habituation than Un-cued (white boxes) in the REM group (Cued REM = black circles and Un-cued REM = gray squares). Thus, a Cueing x Emotion ANOVA on REM sleep gave the main effect of Cueing (p < 0.02). Follow up t tests showed that this was driven by responses to negative stimuli (p = 0.006), where 12 of the 15 participants showed the effect (see lines joining Cued an Un-cued). A trend (p = 0.12) was also present in neutral stimuli, where 13 out of 15 participants showed this effect, but there was more variability in the amount of habituation that occurred. The same ANOVA on SWS showed no effect of Cueing. Instead, there was a main effect of emotion, with neutral items habituating more than negative items (p = 0.02). Box and whisker plots show the 10th and 90th percentiles. REM group, N = 15, SWS group, N = 18.

significant interaction between Group and Cueing ($F_{(1,32)}$ = 5.341, p = 0.027) suggesting that the impact of cueing on this measure of habituation depends on the cognitive state during which TMR is applied. The 2 × 2 × 2 ANOVA gave no other significant result. To determine which sleep stage drove the Group x Cueing interaction, we then conducted separate 2 × 2 ANOVAs with the factors Cueing and Emotion in SWS and REM groups, respectively. This showed a main effect of cueing in REM ($F_{(1,14)}$ = 7.48, p < 0.02) but not SWS ($F_{(1,18)}$ = 0.086, p = 0.8), see Fig. 2. Closer examination of the REM group showed that the effect of Cueing was driven by the negative items (paired test t = 3.21; p = 0.006), with neutral items showing a trend towards the same effect (t = 1.6; p = 0.102). Interestingly, there was a main effect of emotion in the SWS group ($F_{(1,18)}$ = 6.28, p = 0.022) but not in the REM group. Closer examination showed that this was driven by greater habituation of neutral items compared to negative items in the Un-cued condition (t = 2.25, p = 0.037), with a trend in the Cued condition (t = 1.45, p = 0.163).

Memory for the sound-image pairs was near the ceiling during the pre-sleep test. On average, participants responded correctly on >97% of trials. The mean proportion of incorrect trials was as follows [REM group: 1.45% (±1.98) trials (pre-sleep); 1.55% (±1.83) trials (post-sleep); SWS group: 2.05% (±2.77) trials (pre-sleep); 2.62% (±1.08) trials (post-sleep)]. To test for differences in pre-sleep learning of sound-image which could have biased the results, we conducted ANOVAs on pre-sleep proportion correct memory trials with the factors cued/not cued and negative/neutral for the REM and SWS groups, respectively. This revealed no main effects or interactions in the either REM group or SWS group (p > 0.1 in all cases). To test for impacts of TMR or emotion on overnight changes in memory, we performed the same two ANOVAs, now with the overnight change in the proportion of correct memory trials as the dependent variable. This revealed no significant effects for either SWS (p > 0.5) in all cases or REM (p > 0.8) in all cases.

To ensure that reaction times did not differ markedly between cued and un-cued items during the pre-sleep arousal test, we performed a 2 × 2 ANOVA with the factors valence (neutral, negative) and cueing (un-cued, cued) within each group of participants. This revealed a main effect of emotion (F = 29.9, p < 0.001) in the SWS group, and a trend towards the same effect of emotion (F = 3.984, p = 0.066) in the REM group. Because response times are often modulated by both TMR and emotion, we examined the effects of cueing and valence on overnight change in reaction times for REM and SWS groups. We used a pair of 2 × 2 ANOVA with the factors valence (neutral, negative) and cueing (un-cued, cued). This revealed no main effect or interaction in either REM or SWS groups (p > 0.15) in all cases, see Supplementary Information Table S1 for means.

Sleep stage data for each group are reported in Table 2. Note that REM time, SWS time, N2 time, and N1 time did not differ significantly between groups (p > 0.1 in all cases). However, total sleep time did differ significantly (p = 0.01), being shorter on average in the REM than the SWS group. To determine whether there was a relationship between habituation and time spent in REM or SWS or spectral power in slow wave, delta, theta, and gamma bands we conducted a series of Pearson correlations with data across both REM and SWS groups; these revealed several marginal correlations, but none of these survived correction for our four multiple comparisons (p > 0.05 in all cases).

## Discussion

Our data support a role for REM sleep in overnight habituation of arousal responses to picture-sound combinations by demonstrating that TMR in REM but not SWS leads to increased overnight habituation. This finding is in line with a previous observation that REM TMR of sounds used in pre-sleep Pavlovian conditioning resulted in a greater habituation towards these sounds the next day than TMR of the same sounds in stage 2 sleep[26]. These two studies combine to support the one aspect of

**Table 2 Sleep stage data (means).**

|  | TWT | TST | Stage 1 | Stage 2 | SWS | REM |
|---|---|---|---|---|---|---|
| **REM group** |  |  |  |  |  |  |
| Duration (min) | 32.27 (44.94) | 432.93 (63.30) | 21.80 (7.31) | 237.83 (42.13) | 101.27 (22.65) | 72.03 (26.10) |
| % TST |  |  | 5.03 | 54.93 | 23.39 | 16.64 |
| **SWS group** |  |  |  |  |  |  |
| Duration (min) | 16.08 (2.03) | 475.63 (32.8) | 23.29 (10.03) | 252.18 (47.43) | 116.63 (29.18) | 83.53 (20.72) |
| % TST |  |  | 4.9 | 53.02 | 24.52 | 17.56 |

Duration of time [in minutes and as a percentage of total sleep time (TST)] spent in each sleep stage is shown. *TWT* total wake time, *TST* total sleep time, *SWS* slow-wave sleep, N = 18; *REM* rapid-eye movement sleep, N = 15. Means are shown ± standard deviation.

the sleep to forget, sleep to remember hypothesis[27], e.g., the idea that memory reactivation in REM is specifically associated with reductions in subsequent arousal.

A recent study by Lehmann et al[22]. used a similar design to investigate the role of TMR in REM and SWS in emotional memory consolidation and found no effect of cueing on subjective habituation in any group[22]. One possible reason for the difference between these results and our own may stem from the fact that our participants rated pictures for arousal immediately before and after sleep, while participants in Lehmann 2016 performed learning and retrieval tasks after they rated the stimuli but before sleep. Repeated viewing of stimuli can lead to habituation, so Lehmann's design may have allowed pre-sleep habituation to drown out any effects of TMR.

A number of studies suggest that SWS may play a role in habituation[8], for instance, the observation that aspects of autonomic habituation are predicted by SWS[28], that SWS percentage predicts overnight emotional attenuation[29], and that blocking the release of noradrenaline during SWS diminished this habituation[12]. Interestingly, we observed significantly stronger habituation of neutral as compared to negative items in the SWS group irrespective of TMR cueing, possibly because the neutral items are less strongly arousing, to begin with, and it is, therefore, easier for participants to alter the way these items are rated than it is for strongly arousing negative. Alternatively, the arousal status of neutral items could be more open to interpretation than the arousal status of negative items, for instance, responses to negative pictures may be influenced by top-down processes that identify them as conceptually negative and increase the likelihood of a strong arousal rating. Although difficult to interpret, the marked difference between this pattern of consolidation in the SWS group and the pattern of habituation observed in the Uncued REM group, where negative and neutral items habituated to the same extent, could suggest that REM TMR disrupts a natural process of habituation across NREM sleep which works better for comparatively neutral stimuli than for those which are more arousing.

While our findings appear to support a unique role for REM sleep in modulating emotional arousal, we found no correlation between REM time or theta power and habituation. As REM occurs later in the night than SWS we cannot rule out the possibility that the dissociation we observed between these sleep stages was caused by this difference in timing. Notably, however, in Rihm and Rasch[26] cue-related habituation was more pronounced when cues were presented during REM than stage 2 sleep within the same period of early morning sleep, supporting the idea that this difference depends on the sleep stage in which TMR is performed rather than the time of night at which it is performed.

Only female participants were included in this study. This was due to previous studies reporting a greater self-reported response towards negative stimuli in female vs. male individuals. Future work should extend these investigations to males. Finally, it is

worth noting that, throughout the study, participants rated the sound-picture pair, rather than just pictures or sounds. Our results, therefore, relate to this multimodal pair rather than sounds or images alone.

## Methods

**Participants**. Forty-six healthy female volunteers aged 18 to 27 years were tested. The REM group comprised 20 participants: mean age of 24.67 ± 4.42 (SD); two participants were excluded due to not having enough REM to complete the intended five TMR cycles, and three more were excluded due to a technical problem causing data loss. The SWS group comprised 26 participants, mean age 21.31 ± 3.02; three participants were excluded due to not having enough SWS to complete the intended five TMR cycles, and four more were excluded due to a technical problem causing data loss. Excluded participants were not included in any of our stats or tables, thus the N for analysis was 15 for REM and 18 for SWS. Participants had no history of sleep, psychiatric, or neurological disorders, had normal hearing and normal or corrected-to-normal vision. Participants had moderate to intermediate chronotypes according to the Morningness–Eveningness Questionnaire (MEQ)[30]. Participants were not using any psychologically active medications and agreed to abstain from alcohol and caffeine from 24 h prior to the study start. Written informed consent was obtained from all participants and the study was approved by the local ethics committee. All methods were performed in accordance with UREC guidelines.

**Stimuli**. Stimuli consisted of 60 image-sound pairs, of which 30 were negative and 30 neutral (see Supplementary Information Table S2). Images were selected from the International Affective Picture System (IAPS)[31] converted to grayscale, and matched in luminance and resolution (height = 600 px; width = 800 px) using the SHINE toolbox[32] in MATLAB 2007a. The negative and neutral image sets were significantly different in terms of both mean IAPS valence (negative: 2.19 ± 0.57; neutral: 5.04 ± 0.21; $t(58) = 25.71$; $p < 0.001$) and arousal rating (negative: 5.60 ± 0.75; neutral: 3.06 ± 0.62; $t(58) = 14.36$; $p < 0.001$). Each image was paired with a semantically related sound obtained either from the International Affective Digitized Sounds (IADS) database[33] or freely available internet sources (e.g., freesound.org). Each sound was cut to a length of three seconds and volume was normalized across sounds using Audacity 2.0.6. To ensure that sound-picture combinations did not elicit valence ratings that differed from the IAPS only valence ratings in such a way as to lead to an imbalance in our design, we performed a 2 × 2 ANOVA on the initial ratings of the sound picture pairs. Our factors were emotion (negative/neutral) and cueing (cued / not cued). This revealed the expected main effect of emotion ($p < 0.001$), with negative sound/picture pairs rated significantly more arousing than neutral pairs, but no other main effects or interaction ($p > 0.29$ in all cases).

### Experimental tasks

*Arousal rating task*. Participants viewed 60 emotionally negative and neutral images, each presented once in a pseudorandom order (i.e., no more than two images with the same valence presented consecutively) in order to avoid potential emotional priming, expectation, or habituation effects[34–36]. In each trial, the screen presented a dark gray central fixation cross on a black background for 3 s. The 800 (width) × 600 (height) image was then presented in the middle of the screen for 1 s. After this, the fixation screen re-appeared for a further 5 s. A semantically related sound was presented through Sennheiser HD 202 headphones for 3 s starting when the image appeared on the screen. Participants rated each image-sound pair for emotional arousal on a scale of one to nine, using the self-assessment manikin (SAM) scale as a reference[37]. Transition to the next trial was self-paced. The arousal rating task was implemented in Experiment Builder 1.10.1241.

*Sound-image pairing task*. To strengthen the association between each image and its respective sound, participants heard each sound in a pseudorandom order and were subsequently shown two images of the same valence (e.g., both negative) on

either side of the screen. Participants were instructed to select the image previously associated with the sound as quickly and accurately as possible while focusing on a fixation cross displayed in the center of the screen between the two images. Each sound was presented just once.

**PSG & TMR**. Polysomnography (PSG) was recorded using the Embla N7000 recording system, digitally sampled at 500 Hz, and visualized using RemLogic 1.1 software. Electrodes were placed according to the international 10–20 system to six standard scalp locations (F3, F4, C3, C4, O1, and O2), left and right outer canthi, and three chin electrodes. Sleep was scored according to the AASM manual[38].

Half the negative and half the neutral sounds (*n* = 30 total) were presented during REM or SWS, respectively. Sounds have presented a total of five times each in pseudorandom order through PC speakers (Dell A425) placed under the bed. All sounds were integrated into unobtrusive background brown noise (~40 dB overall sound-pressure level), which was presented throughout the entire overnight sleep period. TMR during sleep was manually initiated whenever 30 s of stable REM or SWS were detected in the EEG and paused whenever a movement, a transition to another sleep stage, or visible alpha activity occurred. Sounds were separated by three seconds. Cued and Un-cued sounds were matched according to arousal based on participant-specific subjective arousal ratings immediately prior to sleep. A second researcher confirmed offline that cues were presented exclusively in REM and SWS, in the REM and SWS groups respectively, thus we are confident that all cues were presented in the intended sleep stage.

**Spectral analysis**. Power spectral density (PSD) was analyzed over REM and SWS separately using Welch's method, with power averaged over each time series and across all six channels. This used a 4-s Hamming window length with 50% overlap, focusing on oscillations within the slow wave (0.3–1 Hz) and delta (0.5–4 Hz) in SWS and within theta (4–7 Hz) and gamma (30–40 Hz) frequency bands in REM sleep. Following[8] frontal EEG channels (F3 and F4) were combined for the gamma analysis. Values diverting >2 SD from the mean were excluded.

**Statistics and reproducibility**. Data were analyzed in Jasp, but Linear Mixed Effects analyses were performed in R. Data were analyzed using repeated measures ANOVAs, *t* tests, and Pearson correlations as specified. Results were considered significant at *p* = 0.05. Some participants were excluded due to insufficient sleep, which did not allow for the full TMR manipulation, thus or final sample sizes were 15 for the REM group and 18 for the SWS group.

## Data availability

This study uses behavioral data and sleeps data. Data are available for download using this link:

https://cf-my.sharepoint.com/:f:/g/personal/lewisp8_cardiff_ac_uk/EsxFIMJLyJNGqLMBaR8h4hEBmJZb21hmceCqkY3LCWY7PQ?e=uXgibk.

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

## Acknowledgements

The authors would like to thank Marleen Kempkes for her assistance in creating the stimuli. This project was funded by a BBSRC (BB/J014478/1) Doctoral Training Partnership (DTP) studentship awarded to I.C.H., P.A.L. and G.P. P.A.L. and M.E.A.A. are supported by the ERC grant SolutionSleep 618607.

## Author contributions

I.C.H. designed the experiment, collected the data, analyzed the data, and wrote the manuscript. S.P. and M.-E.T. assisted in design and data collection. M.E.A.A. assisted in scripting and analysis. P.A.L. supervised the project, designed the experiment, analyzed the data, and wrote the paper. G.P. assisted in design of the task, analysis and interpretation. J.H. assisted in analysis and interpretation.

## Competing interests

The authors declare no competing interests.
