## [Peer Review File · Communications Biology]

Reviewers' comments:

Reviewer #1 (Remarks to the Author):

Hutchison and colleagues investigate the role of memory reactivation for attenuating the affective arousal associated with recently formed emotional memory. Specifically, they use targeted memory reactivation - a procedure thought to instigate offline reactivation of the targeted memory - during distinct sleep states (SWS and REM) and assess the effect this has on post-sleep affective arousal. The authors find that TMR during REM but not SWS leads to reduced arousal. This supports an influential hypothesis that memory reactivations during sleep may support the habituation of affective responses for emotional memories and points to a central role for REM sleep in facilitating this process.

The results described are interesting and advance our understanding of the role of sleep and different sleep states in learning and memory. I think these results would be of interest to a broad sleep research audience. I do have a few comments on their analysis which the authors should be able to address easily. If the authors address these then I think the conclusions of their paper would be strengthened considerably.

- The habituation measure: To measure changes in affective arousal following TMR during sleep the authors subtract pre and post sleep arousal ratings and then divide by pre arousal rating. Positive values on this measure would suggest reduction in arousal. However, the pre sleep rating in the denominator is not the same as the pre sleep rating in the numerator (there were two pre sleep ratings). This makes it hard to interpret the habituation measure. The authors should use the same pre sleep (baseline) rating in the numerator and denominator. This could be the first or second pre sleep rating or a combination of the two.

- The authors say the subjects did not recall hearing any of the sounds during the TMR procedure during sleep. However, it would better to measure quantitatively that the TMR did not affect the subjects sleep patterns. For example, as a minimum, to show the total sleep did not differ between the SWS and REM groups, nor the amount of SWS and REM. This would help alleviate the concern that the arousal-reducing effects of REM TMR is not due to some side effect of doing TMR during REM sleep (for example, the subject waking up). It would also be informative (although not essential) to see an analysis of the EEG power spectra during TMR, and compare this to baseline.

Minor comments:

- Fig2 - not clear which group comparisons are significant

Reviewer #2 (Remarks to the Author):

General Comment:

This study from Hutchison and colleagues explores the impact of "reactivating" emotional memories in two separate phases of sleep: NREM and REM sleep. Indeed, while REM sleep is often presented as playing a crucial role in individuals' emotional balance, the literature is not clear regarding the respective role of REM and NREM sleep in the regulation of emotional responses. The present study directly addresses this question and brings interesting novel results. The methodology seems sound, the results are presented clearly and their interpretations supported by the reported data. I have

therefore only minor comments.

Comments:

Introduction

1. The authors wrote in the introduction that “REM is even associated with reduced responses in the amygdala (van der Helm et al., 2011)” but several studies seem to suggest great activations in the amygdala in REM vs Wake (Maquet et al. Nature, 1996; Nofzinger et al. Brain Research 1997 and Braun et al. Science 1998). Perhaps, a better definition of what the authors mean by “reduced responses” would clarify this.

Results:

2. Can the authors clearly state that Slow Wave Sleep was defined as NREM Stage 3? There is no such thing as SWS in the AASM guidelines referenced by the authors in the methods. I agree though that the term SWS is better for the title and main text.

3. Can the authors report the number of times, REM-TMR and SWS-TMR sounds were played overnight (mean and SD across participants)? Was there any difference? Is the number of presentations correlated with the habituation effects reported?

4. Can the authors also report the timing (from sleep onset) of these presentations? Was the timing predictive of the habituation? Was the timing a better predictor than the sleep stage in which sounds were played (SWS vs. REM)?

5. The formula to compute the overnight habituation is “second pre-sleep arousal rating – post-sleep arousal rating) / first pre-sleep arousal rating”. Why using both the first and second pre-sleep arousal ratings?

6. “we conducted a series of Pearson correlations with data across both REM and SWS groups”. Why not using linear mixed-effects models and examine all these predictors together? This could be better as they are likely auto-correlated? The authors could add the number of presentations for each item as a predictor (see comment 3). A similar LME approach could be used when examining habituation scores.

Discussion:

7. The authors suggest that there could be “a natural process of habituation across NREM sleep which works better for comparatively neutral stimuli than for those which are more arousing.” Could this be related to the idea that memories are down-scaled during NREM sleep (Synaptic Homeostasis Hypothesis for example)?

Methods:

8. What was the sound level of the stimuli played overnight? Only the level of the background noise seems to be mentioned.

9. “Each image was paired with a semantically related sound”. Can the authors clarify this and provide examples? Or even better, provide the materials as supplementary files?

10. “participants heard each sound in pseudorandom order and were subsequently shown two images of the same valence (e.g. both negative)” From where the valences values came from? The IAPS valence? If I remember correctly, the IAPS database provides rating for both female and male scorers. Given the all-female sample, which valences were used by the authors?

11. “TMR during sleep was manually initiated whenever thirty seconds of stable REM or SWS were detected in the EEG and paused whenever a movement, a transition to another sleep stage, or visible alpha activity occurred.” How many times, for the REM and SWS sessions, were the TMR sounds paused because of an awakening? Were the corresponding items used in the analyses? Could

they have impacted the results?

12. "A second researcher confirmed offline that cues were presented exclusively in REM and SWS, in the REM and SWS groups respectively." Should this be interpreted as the fact that all the cues were presented in the stage in which they were supposed to?

13. Why focusing on the slow-oscillations bands ("slow (0.3–1 Hz) frequency band") and not the delta band ([0.5-4] Hz)?

14. When extracting PSD, did the authors normalised the power spectra? If so, how? If not, I would recommend doing so (e.g. dividing the value in each band by the value of the power between 0.3 and 40Hz).

15. Did the authors extract PSD over all REM and SWS epochs? Over the 30s windows during which stimuli were presented? During stimulus presentations?

Reviewer #3 (Remarks to the Author):

Brief summary of the manuscript

In this manuscript, Hutchison et al perform targeted memory reactivation (TMR) during sleep using the sounds from a previously learnt sound-image association pairs in order to assess TMR-induced changes in arousal levels. This protocol allows them to show that TMR decreases arousal specifically for negative pairs and only if it is performed during REM sleep but not SWS. These results support a specific role for the regulation of emotional memories by REM sleep.

Overall impression of the work

The authors describe here an interesting experiment that adds to the growing body of evidence linking sleep to the regulation of emotional memories. In particular, the use of TMR is an innovative way to investigate these questions and so far only a few studies have applied it specifically to the question of emotional memories.

Some details of the experimental design however make the link between their results and previous results, in particular the Sleep To Remember Sleep to Forget hypothesis (SRSF) unclear. The authors motivate their experiment as a test of this hypothesis. However this hypothesis is focussed on episodic memory. It is unclear whether episodic memory is involved in their task and more generally what kind of memory is being consolidated. Moreover, this addresses only half of the hypothesis since it also states that REM sleep participated in consolidating relevant information from the emotional memory. Although the authors tested for the memory of sound-image pairing before TMR, they did not test after, so we cannot conclude about this aspect of the SRSF hypothesis. Most of my more specific comments aim at trying to get a better picture of how the experiment which yields convincing results is related to these concepts. Overall, I think the authors should be a little more careful in presenting their results as direct support for the Sleep To Remember Sleep to Forget hypothesis.

Specific comments, with recommendations for addressing each comment

1. The study is presented explicitly as a test of the Sleep To Remember Sleep to Forget hypothesis. This hypothesis postulates that the affective tone of a memory is diminished by REM sleep reactivation and the hypothesis focusses on episodic memory. It is not clear what kind of memory is being manipulated by the procedure in the manuscript since there is no learning of the emotional valence of the imaged. Trying to align the protocol with an episodic memory framework, it might suggest that that during the post-sleep arousal rating task the subjects are remembering the pre-

sleep image presentations and that the memory of this event has a reduced affective charge. Is this what the authors are proposing? The exact link between the SRSF hypothesis and the results reported here should be made clearer and accordingly the degree of support provided by their results discussed.

2. In a similar vein, the authors refer to the reduced arousal after sleep as habituation. The use of the term is ambiguous. Two possible interpretations arise : 1. That TMR induces an emotional habituation to the sound-image pair. 2. That TMR induces a consolidation of the habituation process already having taken place during the two presentations of the sound-image pair during sleep. Could the authors clarify their meaning?

3. A follow-up of 2. The second option given above depends on whether there is habituation because of the encoding task. The authors say in the legend of Figure1 that “ratings may have changed as a result of the encoding task”. Was this in fact the case based on their data?

4. Subjects rated the arousal level of the image-sound pair. Whereas the images were selected based on the IAPS in order to obtain half neutral and half negative images, the sounds were matched according to “semantic content”. It is not therefore clear whether the sounds had the same levels of arousal and / or valence as their associated images. The emotional valence/arousal of the sounds should be described in the materials and methods to make the experiment fully reproducible by others.

5. Follow up on point 4. It is possible that the difference between neutral and negative pairs is driven by the valence/arousal of the memory (ie the sound-image pair) or of the valence/arousal of the cue (sound) used for TMR. This should perhaps be commented on and pointed out.

6. Could the authors provide results of the pre-sleep sound-image pairing task? In particular, was pairing more successfully learned for neutral rather than negative image-sound pairs? This could potentially influence the efficacy of TMR.

7. During the arousal rating task, subjects provided their rating for arousal in a self-paced manner. Did the authors find any change in reaction time? This parameter could also be affected given that sleep deprivation enhances impulsiveness[1], and moreover a change in reaction time (albeit in a memory task) has previously been observed during TMR[2].

8. The sound groups used differed both in arousal and valence, but the authors focus exclusively on arousal evaluation. If there was a reason from previous studies to justify this choice, it should be mentioned.

Minor comments

1. There seems to be a small typo in table 1, some SEM are negative and do not appear in parenthesis

2. Table 2 : for neutral, cued first exposure +/- has been misprinted

3. The ref for Lehman et al has Nat Publ Grp as journal instead of Scientific reports

4. I think it is implied in the text, but could the authors specify if indeed the 2-back task used stimuli with no overlap with the images from the sound-image pairs

References

1. Anderson C, Platten CR. Sleep deprivation lowers inhibition and enhances impulsivity to negative stimuli. *Behav Brain Res* [Internet]. 2011;217(2):463–6. Available from: <http://dx.doi.org/10.1016/j.bbr.2010.09.020>
2. Cairney SA, Durrant SJ, Hulleman J, Lewis PA. Targeted Memory Reactivation During Slow Wave Sleep Facilitates Emotional Memory Consolidation. *Sleep*. 2014;37(4):701–7.

Response to reviewers: Many thanks for your thoughtful and constructive comments and suggestions. We have answered these one by one below, with our answers in blue.

Reviewer #1 (Remarks to the Author):

Hutchison and colleagues investigate the role of memory reactivation for attenuating the affective arousal associated with recently formed emotional memory. Specifically, they use targeted memory reactivation - a procedure thought to instigate offline reactivation of the targeted memory - during distinct sleep states (SWS and REM) and assess the effect this has on post-sleep affective arousal. The authors find that TMR during REM but not SWS leads to reduced arousal. This supports an influential hypothesis that memory reactivations during sleep may support the habituation of affective responses for emotional memories and points to a central role for REM sleep in facilitating this process.

The results described are interesting and advance our understanding of the role of sleep and different sleep states in learning and memory. I think these results would be of interest to a broad sleep research audience. I do have a few comments on their analysis which the authors should be able to address easily. If the authors address these then I think the conclusions of their paper would be strengthened considerably.

- The habituation measure: To measure changes in affective arousal following TMR during sleep the authors subtract pre and post sleep arousal ratings and then divide by pre arousal rating. Positive values on this measure would suggest reduction in arousal. However, the pre sleep rating in the denominator is not the same as the pre sleep rating in the numerator (there were two pre sleep ratings). This makes it hard to interpret the habituation measure. The authors should use the same pre sleep (baseline) rating in the numerator and denominator. This could be the first or second pre sleep rating or a combination of the two.

Response: We apologize for not being clearer about our normalization, and the rationale behind it. In designing our experiment, we were aware that repeated exposure to the stimuli during the pre-sleep 'training' phase would lead to habituation, so we collected arousal ratings both before and after this training. We then used the initial rating to normalise both the pre-sleep rating and the post sleep rating in order to get a clean measure of how strong these subsequent ratings were in relation to the original reaction that participants had to the stimuli (giving arousal as a proportion of original rating). To examine overnight habituation, we then subtracted the normalised post sleep measure from the normalised pre-sleep measure.

So our equation was: $[\text{Habituation} = \text{pre-sleep}/\text{initial} - \text{post-sleep}/\text{initial}]$. In the manuscript we expressed this in a shorter form: $[\text{Habituation} = (\text{pre-sleep} - \text{post-sleep}) / \text{initial}]$

Thus, we always normalised by the initial rating. And we did so because we felt this gave us the clearest measure of arousal. Notably, following your comments, we have tried normalising by the pre-sleep rating (ignoring the initial rating entirely). Thus, habituation = $(\text{pre-sleep} - \text{post-sleep})/\text{pre-sleep}$. This gives essentially the same result as our original normalisation. Using this measure, a 2x2 ANOVA with factors 'reactivation' and 'emotion' on the REM group gives a main effect of reactivation ($p=0.046$), with no other significant main effect or interaction. A parallel ANOVA on the SWS group gives a main effect of emotion ($p=0.03$), but no other main effect or interaction.

Although these results are parallel to those we had originally reported, we do not feel this new analysis is as well controlled as our original analysis, which normalised by the initial ratings. We would thus prefer to keep the original analysis, though we will of course change this if you feel that is more appropriate.

REF: The interplay between attention and long-term memory in affective habituation.

Ferrari V, Matria S, Codispoti M. *Psychophysiology*. 2020 Jun;57(6):e13572. doi: 10.1111/psyp.13572. Epub 2020 Apr 2. PMID: 32239721

- The authors say the subjects did not recall hearing any of the sounds during the TMR procedure during sleep. However, it would better to measure quantitatively that the TMR did not affect the subjects sleep patterns. For example, as a minimum, to show the total sleep did not differ between the SWS and REM groups, nor the amount of SWS and REM. This would help alleviate the concern that the arousal-reducing effects of REM TMR is not due to some side effect of doing TMR during REM sleep (for example, the subject waking up). It would also be informative (although not essential) to see an analysis of the EEG power spectra during TMR, and compare this to baseline.

Response: Thanks – we have used independent samples t-test to compare total sleep time, REM time, and SWS time across the two experimental groups (REM replay and SWS replay). REM time, SWS time, N2 time, and N1 time did not differ significantly between groups ($p > 0.1$ in any case). However, total sleep time did differ significantly ($p = 0.01$), being shorter on average in the REM than the SWS group.

We now report this in the article as follows:

‘Sleep stage data for each group are reported in Table 1. Note that REM time, SWS time, N2 time, and N1 time did not differ significantly between groups ($p > 0.1$ in all cases). However, total sleep time did differ significantly ($p = 0.01$), being shorter on average in the REM than the SWS group.’ (pg. 6)

Given that the duration is shorter in the group where TMR was more effective, we feel this is unlikely to explain the efficacy of the TMR. Notably, inclusion of total sleep time as a covariate in our 2x2x2 ANOVA on arousal ratings with factors group, replay, and emotion does not change the result, in that we still see a group*replay interaction ($F = 4.97$, $P = 0.033$), with no other main effect or interaction, and no interaction of any factor with total sleep time.

Unfortunately we did not have EEG triggers to mark when the cues were applied, so we are unable to analyse the spectrum during TMR.

Reviewer #2 (Remarks to the Author):

General

Comment:

This study from Hutchison and colleagues explores the impact of “reactivating” emotional memories in two separate phases of sleep: NREM and REM sleep. Indeed, while REM sleep is often presented as playing a crucial role in individuals’ emotional balance, the literature is not clear regarding the respective role of REM and NREM sleep in the regulation of emotional responses. The present study directly addresses this question and brings interesting novel results. The methodology seems sound, the results are presented clearly and their interpretations supported by the reported data. I have therefore only minor comments.

Comments:

Introduction

1. The authors wrote in the introduction that “REM is even associated with reduced responses in the amygdala (van der Helm et al., 2011)” but several studies seem to suggest great activations in the amygdala in REM vs Wake (Maquet et al. *Nature*, 1996; Nofzinger et al. *Brain Research* 1997 and Braun et al. *Science* 1998). Perhaps, a better definition of what the authors mean by “reduced responses” would clarify this.

Response: Thanks for pointing out our unclear wording. You are correct in saying that the amygdala has been shown to be active in REM. What we intended to say was different from this, we should have said that REM has been linked to overnight reductions in amygdala response to arousing stimuli. We have now amended the text as follows:

'This proposal is supported by the observation that emotional habituation, which can be defined as a reduction in emotional response, occurs across periods of sleep containing REM (Gujar, McDonald, Nishida, & Walker, 2011). REM has also been associated with overnight reductions in the extent to which the amygdala responds to emotional stimuli (van der Helm et al., 2011).' (pg. 3).

Results:

2. Can the authors clearly state that Slow Wave Sleep was defined as NREM Stage 3? There is no such thing as SWS in the AASM guidelines referenced by the authors in the methods. I agree though that the term SWS is better for the title and main text.

Response: The data were scored according to AASM guidelines, thus we are using 'slow wave sleep' to describe N3. We apologize for any confusion.

3. Can the authors report the number of times, REM-TMR and SWS-TMR sounds were played overnight (mean and SD across participants)? Was there any difference? Is the number of presentations correlated with the habituation effects reported?

Response: Sounds were presented a total of 5 times each in pseudorandom order for all participants presented here. The few participants in whom sounds were not presented the full 5x were excluded from the analysis.

4. Can the authors also report the timing (from sleep onset) of these presentations? Was the timing predictive of the habituation? Was the timing a better predictor than the sleep stage in which sounds were played (SWS vs. REM)?

Response: Unfortunately we do not have markers to indicate when the TMR cues were played, so we are unable to check this.

5. The formula to compute the overnight habituation is "second pre-sleep arousal rating – post-sleep arousal rating) / first pre-sleep arousal rating". Why using both the first and second pre-sleep arousal ratings?

Response: We apologize for not being clearer about our normalization, and the rationale behind it. In designing our experiment, we were aware that repeated exposure to the stimuli during the pre-sleep 'training' phase would lead to habituation, so we collected arousal ratings both before and after this training. We then used the initial rating to normalise both the pre-sleep rating and the post sleep rating in order to get a clean measure of how strong these subsequent ratings were in relation to the original reaction that participants had to the stimuli (giving arousal as a proportion of original rating). To examine overnight habituation, we then subtracted the normalised post sleep measure from the normalised pre-sleep measure.

So our equation was: [Habituation = pre-sleep/initial – post-sleep/initial]. In the manuscript we expressed this in a shorter form: [Habituation = (pre-sleep – post-sleep) / initial]

Thus, we always normalised by the initial rating. And we did so because we felt this gave us the clearest measure of arousal. Notably, following your comments, we have tried normalising by the pre-sleep rating (ignoring the initial rating entirely). Thus, habituation = (pre-sleep - post-sleep)/pre-sleep. This gives essentially the same result as our original normalisation.

Using this measure, a 2x2 ANOVA with factors 'reactivation' and 'emotion' on the REM group gives a main effect of reactivation ($p=0.046$), with no other significant main effect or interaction. A parallel ANOVA on the SWS group gives a main effect of emotion ($p=0.03$), but no other main effect or interaction.

Although these results are parallel to those we had originally reported, we do not feel this new analysis is as well controlled as our original analysis, which normalised by the initial ratings. We would thus prefer to keep the original analysis, though we will of course change this if you feel that is more appropriate.

REF: The interplay between attention and long-term memory in affective habituation.
Ferrari V, Mastria S, Codispoti M. *Psychophysiology*. 2020 Jun;57(6):e13572. doi: 10.1111/psyp.13572. Epub 2020 Apr 2. PMID: 32239721

6. "we conducted a series of Pearson correlations with data across both REM and SWS groups". Why not using linear mixed-effects models and examine all these predictors together? This could be better as they are likely auto-correlated? The authors could add the number of presentations for each item as a predictor (seem comment 3). A similar LME approach could be used when examining habituation scores.

Response: Thanks for this suggestion. We have now used LME analysis to examine the modulation of overnight habituation by REM, SWS, delta power, theta power, gamma power, or any combination of these. Our new text about this reads as follows:

'To determine whether there was a relationship between habituation and time spent in REM or SWS or spectral power in delta, theta, and gamma bands we conducted a linear mixed effects (LME) analysis. Thus, we used the sleep parameters as continuous covariates to the fixed factors of our LME on habituation. We created a separate model for every combination of these five variables. We then calculated the AIC for all of these models, and compared it to the AIC For the null model (which did not include these variables). We then used an ANOVA to compare the model having the lowest AIC to the null model. This gave a non-significant p value ($P=0.07$), we thus conclude that none of these covariates significantly modulates habituation.' (bottom of pg. 6).

We did not include number of sound presentations as a covariate, since this number was always 5 (participants in whom all sounds were not presented the full 5 times were excluded from the analysis).

Discussion:

7. The authors suggest that there could be "a natural process of habituation across NREM sleep which works better for comparatively neutral stimuli than for those which are more arousing." Could this be related to the idea that memories are down-scaled during NREM sleep (Synaptic Homeostatis Hypothesis for example)?

Response: We agree that this part of the discussion, though speculative, was a bit underdeveloped. We believe that the ratings of neutral items, which are less strongly arousing to begin with, may be more easily influenced by sleep or other manipulations than strongly arousing negative items. We have now explained this idea as follows:

'Interestingly, we observed significantly stronger habituation of neutral as compared to negative items in the SWS group irrespective of TMR cueing, possibly because the neutral items are less strongly arousing to begin with, and it is therefore easier for participants to alter the way these items are rated than it is for strongly arousing negative.' (pg. 9).

Having said this, is not clear to us that the synaptic homeostasis hypothesis, which relates to generalised downscaling of synapses across the brain, would relate specifically to reductions in arousal, so we have not chosen to discuss that possibility here.

Methods:

8. What was the sound level of the stimuli played overnight? Only the level of the background noise seems to be mentioned.

Response: Sounds were presented through PC speakers (Dell A425) placed under the bed. Although we did not record the exact intensity of sound presentation, all sounds were integrated into unobtrusive background brown noise (approximately 40dB overall sound-pressure level), which was presented throughout the entire overnight sleep period.

We now specify this as follows: ‘Sounds were presented a total of 5 times each in pseudorandom order through PC speakers (Dell A425) placed under the bed. All sounds were integrated into unobtrusive background brown noise (approximately 40dB overall sound-pressure level), which was presented throughout the entire overnight sleep period’ (pg. 13)

9. “Each image was paired with a semantically related sound”. Can the authors clarify this and provide examples? Or even better, provide the materials as supplementary files?

Response: Thanks for this suggestion. The sounds and pictures came from the IADS and IAPS databases, and can therefore not be published en bulk. However we have now included several pairs as examples in the supplementary material. We also include the below supplementary table with descriptions of the images used in each category.

Supplementary Table S4 Neutral and negative IAPS images

30 neutral and 30 negative images from the International Affective Picture System (IAPS) were selected for this study. Pictures were converted to grey scale and matched in luminance and resolution to minimize light-specific variation in respect to pupil dilation. A description of the content of each of the stimuli is listed separately for negative and neutral images.

Negative IAPS	Neutral IAPS
Pitbull	Tropical bird
Shark	Farmer
Toddler with flies on face	Neutral male face
Grieving female with corpse	Female secretary on phone
Crying child	Couple walking down stairs
Car accident corpse	Sleeping males on train
Male corpse on train	Two females chatting
Battered female	Female binge eating
Crying male in hospital	Female tourist

Unconscious male on ventilator	Male blow-drying hair
Disabled child	Towel
Bandaged child	Spoon
Distressed female	Porcelain bowl
Military ground attack	Mug
Female toddler crying at male	Fan
Roach on Pizza	Beer glass
Cemetery	Glass candle stick
Rotting cow carcass	Lightbulb
Power plant	Dice
Toilet soiled with feces	Fork
Toilet filled with vomit	Book
Sliced Hand	Truck
Crying soldier	Bus
Male threatened at gun point	Bedside lamp
Crime scene with body	Carpet
Human skulls	Clothes rack
Dead soaked cat	Plate with duck pattern
Burning jet	House
Filled ashtray	Traffic jam
Car Accident	Box of tissues

10. "participants heard each sound in pseudorandom order and were subsequently shown two images of the same valence (e.g. both negative)" From where the valences values came from? The IAPS valence? If I remember correctly, the IAPS database provides rating for both female and male scorers. Given the all-female sample, which valences were used by the authors?

Response: We used the IAPS valence ratings from 'all subjects' to determine which images were categorised as negative and which neutral.

11. "TMR during sleep was manually initiated whenever thirty seconds of stable REM or SWS were detected in the EEG and paused whenever a movement, a transition to another sleep stage, or visible alpha activity occurred." How many times, for the REM and SWS sessions, were the TMR sounds paused because of an awakening? Were the corresponding items used in the analyses? Could they have impacted the results?

Response: None of the participants included in our analysis were awake during a sound at any point (i.e. no alpha). The experimenter carefully monitored this the PSG during stimulation, and paused the TMR before the sound started if she noticed they were slipping out of stable REM or SWS.

12. “A second researcher confirmed offline that cues were presented exclusively in REM and SWS, in the REM and SWS groups respectively.” Should this be interpreted as the fact that all the cues were presented in the stage in which they were supposed to?

Response: Yes, that’s right, all cues were presented in the stage intended. We have now clarified as below:

‘A second researcher confirmed offline that cues were presented exclusively in REM and SWS, in the REM and SWS groups respectively, thus we are confident that all cues were presented in the intended sleep stage.’ Pg. 13

13. Why focusing on the slow-oscillations bands (“slow (0.3–1 Hz) frequency band”) and not the delta band ([0.5-4] Hz).

Response: This is a reasonable query. We have now extended our analysis to include the delta band. This does not change the results, but at least it is more comprehensive. The new text reads as follows:

‘To determine whether there was a relationship between habituation and time spent in REM or SWS or spectral power in slow wave, delta, theta, and gamma bands we conducted a series of Pearson correlations with data across both REM and SWS groups; these revealed no significant correlation ($p > 0.05$ in all cases).’ (pg. 6)

14. When extracting PSD, did the authors / the power spectra? If so, how? If not, I would recommend doing so (e.g. dividing the value in each band by the value of the power between 0.3 and 40Hz).

Response: We had not previously normalised in this way done so and re-run these correlations. This does not change our results, so we kept the original version.

15. Did the authors extract PSD over all REM and SWS epochs? Over the 30s windows during which stimuli were presented? During stimulus presentations?

Response: PSD were extracted from the full REM and SWS epochs.

Reviewer #3 (Remarks to the Author):

Brief summary of the manuscript

In this manuscript, Hutchison et al perform targeted memory reactivation (TMR) during sleep using the sounds from a previously learnt sound-image association pairs in order to assess TMR-induced changes in arousal levels. This protocol allows them to show that TMR decreases arousal specifically for negative pairs and only if it is performed during REM sleep but not SWS. These results support a specific role for the regulation of emotional memories by REM sleep.

Overall impression of the work

The authors describe here an interesting experiment that adds to the growing body of evidence linking sleep to the regulation of emotional memories. In particular, the use of TMR is an innovative way to investigate these questions and so far only a few studies have

applied it specifically to the question of emotional memories.

Some details of the experimental design however make the link between their results and previous results, in particular the Sleep To Remember Sleep to Forget hypothesis (SRSF) unclear. The authors motivate their experiment as a test of this hypothesis. However this hypothesis is focussed on episodic memory. It is unclear whether episodic memory is involved in their task and more generally what kind of memory is being consolidated. Moreover, this addresses only half of the hypothesis since it also states that REM sleep participated in consolidating relevant information from the emotional memory. Although the authors tested for the memory of sound-image pairing before TMR, they did not test after, so we cannot conclude about this aspect of the SRSF hypothesis. Most of my more specific comments aim at trying to get a better picture of how the experiment which yields convincing results is related to these concepts. Overall, I think the authors should be a little more careful in presenting their results as direct support for the Sleep To Remember Sleep to Forget hypothesis.

Response: We appreciate this input and take the reviewers point in that, as our paper does not deal with memory but instead focuses only on arousal ratings and habituation of these across sleep, we can only really address one half of the SFSR hypothesis.

We have altered the manuscript throughout to clarify this point (see below):

Abstract:

These results support one aspect of the sleep to forget, sleep to remember (SFSR) hypothesis which proposes that emotional memory reactivation during REM sleep underlies sleep-dependent habituation.

Introduction:

In the sleep to forget, sleep to remember (SFSR) hypothesis, Walker and colleagues propose that this dissipation of emotional charge relies on reactivation of the emotional memory during rapid-eye movement (REM) sleep (Goldstein and Walker, 2014; van der Helm et al., 2011).

...

'We set out to test the SFSR prediction that replaying emotionally arousing memories during REM, but not SWS, would be associated with a reduced arousal ratings for this material the next day. We did this by manipulating the reactivation of emotional memories during sleep using TMR.' (pg. 4)

...

'Based on the SFSR hypothesis, we predicted that TMR would lead to greater habituation when applied during REM, but not SWS.' (pg. 4)

Discussion:

'These two studies combine to support the one aspect of the sleep to forget, sleep to remember hypothesis (Walker et al., 2009), e.g. the idea that memory reactivation in REM is specifically associated with reductions in subsequent arousal.' (pg. 9)

Specific comments, with recommendations for addressing each comment

1. The study is presented explicitly as a test of the Sleep To Remember Sleep to Forget hypothesis. This hypothesis postulates that the affective tone of a memory is diminished by REM sleep reactivation and the hypothesis focusses on episodic memory. It is not clear what kind of memory is being manipulated by the procedure in the manuscript since there is no learning of the emotional valence of the imaged. Trying to align the protocol with an episodic

memory framework, it might suggest that that during the post-sleep arousal rating task the subjects are remembering the pre-sleep image presentations and that the memory of this event has a reduced affective charge. Is this what the authors are proposing? The exact link between the SFSF hypothesis and the results reported here should be made clearer and accordingly the degree of support provided by their results discussed.

Response: The reviewer correctly states that the SFSR hypothesis postulates that the affective tone of a memory is diminished by REM sleep reactivation. The hypothesis also proposes that memory reactivation in REM leads to a reduction in arousal due to the absence of norepinephrine, and thus autonomic responses in REM (Goldstein & Walker, 2014; van der Helm et al., 2011). Van der Helm and Walker talk about a ‘decoupling’ of emotion from memory, since the memory can continue to consolidate and be maintained while the emotional content fades away. In our current study, we set out to test this aspect of the hypothesis – namely, whether reactivating the memory in REM, as compared to SWS (which is not associated with drastic norepinephrine reductions) would lead to habituation. We did this by applying TMR in SWS and in REM, and looking at the differential impacts of this manipulation upon arousal ratings. And our results support the idea that TMR cued memory reactivation in REM is associated with a reduction in arousal ratings, while TMR cued memory reactivation in SWS is not.

The reviewer points out that the hypothesis also involved ideas about what would happen to the memory during reactivation in REM, namely, the idea that the memory itself might be strengthened or at least protected even as the emotional content was decoupled from it. Unfortunately we were not able to test element of the hypothesis, since our study focussed exclusively on arousal ratings. We have now taken much more care to clarify which aspect of the SFSR hypothesis we are testing, and how our data support it, please see below:

Abstract:

‘Our results show that TMR during REM, but not SWS significantly decreased subjective arousal, and this effect is driven by the more negative stimuli. These results support one aspect of the sleep to forget, sleep to remember (SFSR) hypothesis which proposes that emotional memory reactivation during REM sleep underlies sleep-dependent habituation.’

Introduction

‘..the emotional charge associated with these memories is gradually lost over time (Dolcos, LaBar, & Cabeza, 2005). In the sleep to forget, sleep to remember (SFSR) hypothesis, Walker and colleagues propose that this dissipation of emotional charge relies on reactivation of the emotional memory during rapid-eye movement (REM) sleep (Goldstein & Walker, 2014; van der Helm et al., 2011). This proposal is supported by the observation that emotional habituation, which can be defined as a reduction in emotional response, occurs across periods of sleep containing REM (Gujar et al., 2011). REM has also been associated with overnight reductions in the extent to which the amygdala responds to emotional stimuli (van der Helm et al., 2011). However, contrasting results suggest that emotional reactivity is maintained across REM (Baran, Pace-Schott, Ericson, & Spencer, 2012; S Groch, Wilhelm, Diekelmann, & Born, 2012)..’ (pg. 3).

‘Building on this literature, we set out to test the SFSR prediction that replaying emotionally arousing memories during REM, but not SWS, would be associated with a reduced arousal ratings for this material the next day. We did this by manipulating the reactivation of emotional memories during sleep using TMR. Our participants rated emotionally negative and neutral picture-sound pairs for arousal both before and after a night of sleep. During sleep, we cued half of the negative and half of the neutral stimuli for reactivation by softly replaying the associated sounds. We then examined the impact of this TMR upon overnight habituation of the arousal response. We carefully controlled the sleep stage in which TMR was applied,

cueing participants either in REM (REM Group) or SWS (SWS Group). Based on the SFSS hypothesis, we predicted that TMR would lead to greater habituation when applied during REM, but not SWS.’ (pg. 4)

Discussion:

‘Our data support a role for REM sleep in overnight habituation of arousal responses to picture-sound combinations by demonstrating that TMR in REM but not SWS leads to increased overnight habituation. This finding is in line with a previous observation that REM TMR of sounds used in pre-sleep Pavlovian conditioning resulted in a greater habituation towards these sounds the next day than TMR of the same sounds in stage 2 sleep (Rihm & Rasch, 2015). These two studies combine to support the one aspect of the sleep to forget, sleep to remember hypothesis (Walker, van der, & van der Helm, 2009), e.g. the idea that memory reactivation in REM is specifically associated with reductions in subsequent arousal.’ (pg. 9)

2. In a similar vein, the authors refer to the reduced arousal after sleep as habituation. The use of the term is ambiguous. Two possible interpretations arise : 1. That TMR induces an emotional habituation to the sound-image pair. 2. That TMR induces a consolidation of the habituation process already having taken place during the two presentations of the sound-image pair during sleep. Could the authors clarify their meaning?

Response: This is an interesting question, but we should clarify that we are not examining sleep’s impact on memory in this study. Instead, we are examining its impact on emotions. By habituation we therefore mean ‘reduction in arousal response’. While, as the reviewer points out, it is possible that such reductions could be linked to memory consolidation, that is not something we have examined or attempted to discuss. We have now clarified our definition of habituation in the introduction as follows:

‘This proposal is supported by the observation that emotional habituation, defined as a reduction in emotional response, occurs when REM is present (Gujar et al., 2011), and REM is even associated with overnight reductions in the extent to which the amygdala responds to emotional stimuli (van der Helm et al., 2011).’ Pg. 3.

3. A follow-up of 2. The second option given above depends on whether there is habituation because of the encoding task. The authors say in the legend of Figure1 that “ratings may have changed as a result of the encoding task”. Was this in fact the case based on their data?

Response: We tested this by calculating change from initial to pre-sleep tests in negative and neutral items for each group, and comparing this to zero using one-way t-tests. This revealed no significant change ($p > 0.05$) in all cases, though there was a trend to a decrease in arousal in the neutral items of the SWS group ($p = 0.07$).

4. Subjects rated the arousal level of the image-sound pair. Whereas the images were selected based on the IAPS in order to obtain half neutral and half negative images, the sounds were matched according to “semantic content”. It is not therefore clear whether the sounds had the same levels of arousal and / or valence as their associated images. The emotional valence/arousal of the sounds should be described in the materials and methods to make the experiment fully reproducible by others.

Response: We agree with this comment and have now placed sample picture/sound pairs in the supplement. Many of the sounds came from the IADS database, but some came from other online sources, and thus do not have valence/arousal ratings. Thus, instead of providing the latter for the sounds themselves, we have examined the arousal ratings for the sound-

picture combinations. To ensure that these combos did not elicit valence rating that differed from the IAPS only valence ratings in such a way as to lead to an imbalance between the negative and neutral categories in our design, we performed a 2x2 ANOVA on the initial ratings of the sound picture pairs. Our factors were emotion (negative/neutral) and cueing (cued / not cued). This revealed the expected main effect of emotion ($p < 0.001$), with negative sound/picture pairs rated significantly more arousing than neutral pairs, but no other main effects or interaction ($p > 0.29$ in all cases).

We have now included this information in the manuscript as follows:

'To ensure that sound-picture combinations did not elicit valence rating that differed from the IAPS only valence ratings in such a way as to lead to an imbalance in our design, we performed a 2x2 ANOVA on the initial ratings of the sound picture pairs. Our factors were emotion (negative/neutral) and cueing (cued / not cued). This revealed the expected main effect of emotion ($p < 0.001$), with negative sound/picture pairs rated significantly more arousing than neutral pairs, but no other main effects or interaction ($p > 0.29$ in all cases).' Pg. 12

5. Follow up on point 4. It is possible that the difference between neutral and negative pairs is driven by the valence/arousal of the memory (ie the sound-image pair) or of the valence/arousal of the cue (sound) used for TMR. This should perhaps be commented on and pointed out.

Response: This is a fair point, we have added the below sentence to make sure it is clear that all ratings are for sound/picture pairs.

'It is also worth noting that, throughout the study, participants rated the sound-picture pair, rather than just pictures or sounds. Our results therefore relate to this multimodal pair rather than sounds or images alone.' (pg. 10).

6. Could the authors provide results of the pre-sleep sound-image pairing task? In particular, was pairing more successfully learned for neutral rather than negative image-sound pairs? This could potentially influence the efficacy of TMR.

Response: To examine pre-sleep accuracy on the sound-picture memory task, we conducted ANOVAs with the factors cued/not cued and negative/neutral for the REM and SWS groups respectively. This revealed no main effects or interactions in the either REM group or SWS group ($p > 0.1$ in all cases).

It is also worth noting that memory performance was near ceiling. On average, participants responded correctly on >97% of trials. The mean number of incorrect trials was as follows [REM group: 1.45% (± 1.98) trials (pre-sleep); 1.55% (± 1.83) trials (post-sleep); SWS group: 2.05% (± 2.77) trials (pre-sleep); 2.62% (± 1.08) trials (post-sleep)]. We have included this information in the results section as follows:

'Memory for the sound-image pairs was near ceiling during the pre-sleep test. On average, participants responded correctly on >97% of trials. The mean proportion of incorrect trials was as follows [REM group: 1.45% (± 1.98) trials (pre-sleep); 1.55% (± 1.83) trials (post-sleep); SWS group: 2.05% (± 2.77) trials (pre-sleep); 2.62% (± 1.08) trials (post-sleep)]. To test for differences in pre-sleep learning of sound-image which could have biased the results, we conducted ANOVAs on pre-sleep proportion correct memory trials with the factors cued/not cued and negative/neutral for the REM and SWS groups respectively. This revealed no main effects or interactions in the either REM group or SWS group ($p > 0.1$ in all cases). To test for impacts of TMR or emotion on overnight changes in memory, we performed the same two ANOVAs, now with overnight change in proportion of correct memory trials as the dependent

variable. This revealed no significant effects for either SWS ($p>0.5$) in all cases or REM ($p>0.8$) in all cases.' (Pg. 7)

Given these findings, it is unlikely that memory performance affected TMR.

7. During the arousal rating task, subjects provided their rating for arousal in a self-paced manner. Did the authors find any change in reaction time? This parameter could also be affected given that sleep deprivation enhances impulsiveness[1], and moreover a change in reaction time (albeit in a memory task) has previously been observed during TMR[2].

Response: We appreciate this suggestion of examining overnight change in reaction times. We have now performed this analysis, and include it in the manuscript as follows:

'Because response times are often modulated by both TMR and emotion, we examined the effects of cueing and valence on overnight change in reaction times for REM and SWS groups. We used a pair of 2x2 ANOVA with the factors valence (neutral, negative) and cueing (uncued, cued). This analysis revealed no main effect or interaction in either REM or SWS groups ($p>0.15$) in all cases, see Table S1 for means.' (pg. 7)

We have also included the means and standard deviations of reaction times in a new table S1 of the supplement.

8. The sound groups used differed both in arousal and valence, but the authors focus exclusively on arousal evaluation. If there was a reason from previous studies to justify this choice, it should be mentioned.

Response: We chose focus on arousal rather than valence because our own prior studies suggested an effect of TMR on arousal, not valence (unpublished data, PhD thesis of Tia Tsimpanouli, Manchester University, 2017). Additionally, several studies in the literature suggest an effect of TMR on arousal (Sabine Groch et al., 2017; Rihm & Rasch, 2015).

Groch S, Wilhelm I, Diekelmann S, Born J. 2012. The role of REM sleep in the processing of emotional memories: Evidence from behavior and event-related potentials. *Neurobiol Learn Mem*.

Rihm JS, Rasch B. 2015. Replay of conditioned stimuli during late REM and stage N2 sleep influences affective tone rather than emotional memory strength. *Neurobiol Learn Mem* **122**:142–51. doi:10.1016/j.nlm.2015.04.008

Minor comments

1. There seems to be a small typo in table 1, some SEM are negative and do not appear in parenthesis

Response: We apologise, this has been corrected.

2. Table 2 : for neutral, cued first exposure +/- has been misprinted

Response: Thanks, we have fixed this.

3. The ref for Lehman et al has Nat Publ Grp as journal instead of Scientific reports

Response: Thanks, we have corrected this.

4. I think it is implied in the text, but could the authors specify if indeed the 2-back task used stimuli with no overlap with the images from the sound-image pairs

Response: Thanks – we have detailed this as follows:

'Finally, participants performed a visual 2-back task using images that were not seen elsewhere in the experiment to create a buffer between exposure to the highly emotional stimuli and subsequent sleep.'

References

1. Anderson C, Platten CR. Sleep deprivation lowers inhibition and enhances impulsivity to negative stimuli. *Behav Brain Res* [Internet]. 2011;217(2):463–6. Available from: <http://dx.doi.org/10.1016/j.bbr.2010.09.020>
2. Cairney SA, Durrant SJ, Hulleman J, Lewis PA. Targeted Memory Reactivation During Slow Wave Sleep Facilitates Emotional Memory Consolidation. *Sleep*. 2014;37(4):701–7.

REVIEWERS' COMMENTS:

Reviewer #1 (Remarks to the Author):

I am happy with the response of the authors to my suggestions, I feel they have adequately addressed the concerns I raised with the first version of the manuscript. Although they find a significant difference in total sleep time for the two sleep-state groups, the possibility that this confounds their results is addressed in a covariance analysis.

I recommend the paper for publication as is.

Reviewer #2 (Remarks to the Author):

The authors have answered all my comments. I wish to congratulate them for this very interesting work!

Reviewer #3 (Remarks to the Author):

General comments

The authors had made a thorough and extensive review that both clarifies their approach and its relation to the broader context of the literature. I recommend publication in its current form. I have a minor remark concerning the writing of the discussion below.

Just as a point of interest, I enjoyed reading their thoughts on situating their study in the SRSF hypothesis which helped me clarify my thinking on this. I understand that their protocol aimed at addressing the prediction that REM sleep specifically should be involved in reduced emotional arousal. However the hypothesis bares on the emotions associated with a memory specifically – hence the attempts in some articles to compare arousal levels of stimuli seen before and after sleep with stimuli seen only after sleep (therefore rated by independent subjects), for example [1]. I believe that no crystal clear results have yet emerged from this comparison and so it is as yet difficult to conclude on whether the effect of REM sleep is specific to previously formed memories. Given the design of the study, it evaluates the arousal to previously experienced sound-pairs images and this is a great strength since the pairing of stimuli ensures a balanced protocol. However, I believe their novel approach using TMR could allow for example to ask whether the presentation of sounds during REM that had not been previously experienced during the day could also lead to a reduction in their emotional valence (relative to evaluation by other subjects which I imagine makes an effect harder to detect). This might allow to assess the specificity of the effect on arousal to novel or repeated (therefore memorized) stimuli.

Minor points

1. There may be something I am missing but, in the discussion, I have the impression that the two paragraphs quoted below are discussing the same thing. If not could the authors rephrase a little to make the difference between the two sets of ideas clearer? If they do refer to the same effect (increased habituation for neutral stimuli), I would suggest avoiding the term 'consolidation', especially since the authors are focussing on the "arousal side" of SFSR and not the "memory side". « Interestingly, we observed significantly stronger habituation of neutral as compared to negative items in the SWS group irrespective of TMR cueing, possibly because the neutral items are less

strongly arousing to begin with, and it is therefore easier for participants to alter the way these items are rated than it is for strongly arousing negative.”

“It is noteworthy that, in the SWS group where cueing was apparently ineffective, neutral items consolidated more across the night than negative items. This could be due to the fact that the arousal status of neutral items is more open to interpretation than the arousal status of negative items, e.g. responses to negative pictures may be influenced by top-down processes that identify them as conceptually negative and increase the likelihood of a strong arousal rating.”

[1] E.F. Pace-Schott, E. Shepherd, R.M.C. Spencer, M. Marcello, M. Tucker, R.E. Propper, R. Stickgold, *Neurobiol. Learn. Mem.* 95 (2011) 24–36.

Many thanks for these final comments. We have modified the manuscript based on these, and believe it is again improved. See our responses below.

REVIEWERS' COMMENTS:

Reviewer #1 (Remarks to the Author):

I am happy with the response of the authors to my suggestions, I feel they have adequately addressed the concerns I raised with the first version of the manuscript. Although they find a significant difference in total sleep time for the two sleep-state groups, the possibility that this confounds their results is addressed in a covariance analysis.

I recommend the paper for publication as is.

Reviewer #2 (Remarks to the Author):

The authors have answered all my comments. I wish to congratulate them for this very interesting work!

Reviewer #3 (Remarks to the Author):

General comments

The authors had made a thorough and extensive review that both clarifies their approach and its relation to the broader context of the literature. I recommend publication in its current form. I have a minor remark concerning the writing of the discussion below.

Just as a point of interest, I enjoyed reading their thoughts on situating their study in the SRSF hypothesis which helped me clarify my thinking on this. I understand that their protocol aimed at addressing the prediction that REM sleep specifically should be involved in reduced emotional arousal. However the hypothesis bares on the emotions associated with a memory specifically – hence the attempts in some articles to compare arousal levels of stimuli seen before and after sleep with stimuli seen only after sleep (therefore rated by independent subjects), for example [1]. I believe that no crystal clear results have yet emerged from this comparison and so it is as yet difficult to conclude on whether the effect of REM sleep is specific to previously formed memories. Given the design of the study, it evaluates the arousal to previously experienced sound-pairs images and this is a great strength since the pairing of stimuli ensures a balanced protocol. However, I believe their novel approach using TMR could allow for example to ask whether the presentation of sounds during REM that had not been previously experienced during the day could also lead to a reduction in their emotional valence (relative to evaluation by other subjects which I imagine makes an effect harder to detect). This might allow to assess the specificity of the effect on arousal to novel or repeated (therefore memorized) stimuli.

We thank the reviewer for this insightful comment. We agree that our design does not allow us to clearly determine whether REM acts only upon memories that have

been previously learned in the day, or could indeed act upon memories presented for the first time during sleep. Future work will need to investigate this.

Minor points

1. There may be something I am missing but, in the discussion, I have the impression that the two paragraphs quoted below are discussing the same thing. If not could the authors rephrase a little to make the difference between the two sets of ideas clearer? If they do refer to the same effect (increased habituation for neutral stimuli), I would suggest avoiding the term 'consolidation', especially since the authors are focussing on the "arousal side" of SFSR and not the "memory side".

« Interestingly, we observed significantly stronger habituation of neutral as compared to negative items in the SWS group irrespective of TMR cueing, possibly because the neutral items are less strongly arousing to begin with, and it is therefore easier for participants to alter the way these items are rated than it is for strongly arousing negative.»

"It is noteworthy that, in the SWS group where cueing was apparently ineffective, neutral items consolidated more across the night than negative items. This could be due to the fact that the arousal status of neutral items is more open to interpretation than the arousal status of negative items, e.g. responses to negative pictures may be influenced by top-down processes that identify them as conceptually negative and increase the likelihood of a strong arousal rating."

Thanks for pointing out the similarity of these two excerpts. We have now merged this into a single paragraph as follows:

'Interestingly, we observed significantly stronger habituation of neutral as compared to negative items in the SWS group irrespective of TMR cueing, possibly because the neutral items are less strongly arousing to begin with, and it is therefore easier for participants to alter the way these items are rated than it is for strongly arousing negative. Alternatively, the arousal status of neutral items could be more open to interpretation than the arousal status of negative items, for instance responses to negative pictures may be influenced by top-down processes that identify them as conceptually negative and increase the likelihood of a strong arousal rating. Although difficult to interpret, the marked difference between this pattern of consolidation in the SWS group and the pattern of habituation observed in the Un-cued REM group, where negative and neutral items habituated to the same extent, could suggest that REM TMR disrupts a natural process of habituation across NREM sleep which works better for comparatively neutral stimuli than for those which are more arousing.' pg. 9.

[1] E.F. Pace-Schott, E. Shepherd, R.M.C. Spencer, M. Marcello, M. Tucker, R.E. Propper, R. Stickgold, *Neurobiol. Learn. Mem.* 95 (2011) 24–36.

** See Nature Research's author and referees' website at www.nature.com/authors for information about policies, services and author

benefits

Our flexible approach during the COVID-19 pandemic

If you need more time at any stage of the peer-review process, please do let us know. While our systems will continue to remind you of the original timelines, we aim to be as flexible as possible during the current pandemic.

COMMSBIO - This email has been sent through the Springer Nature Tracking System NY-610A-NPG&MTS

Confidentiality Statement:

This e-mail is confidential and subject to copyright. Any unauthorised use or disclosure of its contents is prohibited. If you have received this email in error please notify our Manuscript Tracking System Helpdesk team at <http://platformsupport.nature.com> .

Details of the confidentiality and pre-publicity policy may be found here <http://www.nature.com/authors/policies/confidentiality.html>

Privacy Policy | Update Profile

DISCLAIMER: This e-mail is confidential and should not be used by anyone who is not the original intended recipient. If you have received this e-mail in error please inform the sender and delete it from your mailbox or any other storage mechanism. Springer Nature Limited does not accept liability for any statements made which are clearly the sender's own and not expressly made on behalf of Springer Nature Ltd or one of their agents. Please note that Springer Nature Limited and their agents and affiliates do not accept any responsibility for viruses or malware that may be contained in this e-mail or its attachments and it is your responsibility to scan the e-mail and attachments (if any).

Springer Nature Ltd. Registered office: The Campus, 4 Crinan Street, London, N1 9XW. Registered Number: 00785998 England.

Attachments area